# Temporal and Spatial Coupling Methods for the Efficient Modelling of Dynamic Solids

**DOI:** 10.3390/ma18051080

**Published:** 2025-02-28

**Authors:** Kin Fung Chan, Nicola Bombace, Indrajeet Sahu, Simone Falco, Nik Petrinic

**Affiliations:** 1Department of Engineering Science, University of Oxford, Parks Road, Oxford OX1 3PJ, UK; indrajeet.sahu@eng.ox.ac.uk (I.S.); simone.falco@eng.ox.ac.uk (S.F.);; 2Adaptive, Embedded and AI (AEAI) Group, Advanced Micro Devices Inc., Darwin House, Edinburgh Technopole, Bush Estate, Edinburgh EH26 0PY, UK; nico.bombace@amd.com

**Keywords:** multi-time stepping, non-matching meshes, explicit finite elements, dynamic modelling, heterogeneous discretisations

## Abstract

This paper presents efficient coupling methods that accurately reduce the computational cost for modelling solids dynamically with finite elements. A multi-time-step integration algorithm is developed to leverage varying time steps throughout a domain. Interfaces between subdomains are resolved explicitly with the continuity of acceleration and tractions. A spatial coupling method is combined with multiple time steps, allowing for meshes that do not necessarily conform at their interfaces. The method avoids solving additional degrees of freedom at these interfaces, with parameter-free coupling operators defined between meshes. A speedup >12× is achieved in comparison to reference single-time-step methods.

## 1. Introduction

The temporal and spatial discretisation of structural dynamic problems is directly related to the accuracy and computational cost of the explicit finite element method. Constrained by the Courant–Friedrichs–Lewy (CFL) condition, the critical time step, $\Delta t_{C}$, is proportional to the element size, and inversely proportional to the dilatational wave speed [1]. This leads to simulations being restricted to min{Δt} of the element with the smallest size or highest wave speed. Initially referred to as subcycling, pioneering works allowed for the integration of multiple time steps in a single domain [2,3]. The coupling of various kinematic fields was explored, along with the stability of such algorithms [4,5]. Asynchronous variational integration is a non-trivial alternative, discretising the functional instead of the equations of motion [6]. The main drawback being its high complexity implementation. Heterogeneous asynchronous time integration extended methods to varying, non-integer and large time step ratios [7,8]. However, maintaining the continuity of kinematics at the interface remains a challenge, especially for all three fields [9]. In recent works, energy-conserving methods have been developed; however, they depart far from the CFL condition to resolve the interface conditions [10]. Spatially, non-matching mesh algorithms facilitate more flexible geometric modelling [11]. Nitsche’s method has shown to weakly enforce conditions on the interfaces of non-matching meshes without additional unknowns, but commonly suffers from sensitivity to parameters [12,13,14,15]. Following similar principles of weak continuity, Lagrange multipliers are called upon in the mortar-based methods [16,17,18,19,20]. These types of methods have proven to be very robust, however they struggle to fulfil the inf-sup stability condition, and incur a large computational cost with mapping master and slave nodes [21]. In comparison, the use of localised Lagrange multipliers introduces a frame that independently enforces compatibility constraints on each boundary [22,23,24,25,26,27]. However, as is common with other methods that introduce an additional discretised interface, the construction of this interface is neither trivial nor computationally cheap [28,29,30,31]. This brief review justifies the need for more efficient couplings, both temporally and spatially. Coupling algorithms that allow for varying time-step ratios, whilst stepping close to the CFL condition, concertedly those that solve for non-matching meshes, without increasing the degrees of freedom, remain a hot topic of research.

## 2. Governing Equations of Dynamic Solids

The problem of a solid body subject to impact is described through the partitioning of a domain as illustrated in Figure 1. The deformation is governed by the momentum balance equation acting on the solid domain Ω:(1)$$\begin{array}{c} \rho \ddot{u}=\nabla \cdot \sigma +\rho b,\;\mathrm{in}\;\Omega \times \left[0,T\right] \end{array}$$
where ρ, $\ddot{u}$, σ and b denote the density, acceleration field, Cauchy stress tensor and body forces. Deformation is described at time t∈[0,T] for a specified constant T>0. Updated Lagrangian formulations of a single body are found in the following [32,33,34]; here, we extend this to a partitioned formulation to solve multiple solid subdomains.

Defining multiple subdomains, we state B is a solid body in an open region $\Omega \subset R^{3}$, with its boundary denoted ∂Ω. Ω is partitioned into S non-overlapping subdomains:(2)$$\begin{array}{c} \Omega =\overset{S}{\underset{i=1}{⋃}}\Omega_{i}\;\mathrm{and}\;\Omega_{i}\cap \Omega_{j}=Ø\;\mathrm{for}\;i\neq j \end{array}$$

Starting from S=2, for a two-subdomain partitioning, where $\Omega_{L}$ and $\Omega_{s}$ denote large and small subdomains. On the boundary of the two matching subdomains, Γ, we enforce:(3)$$\begin{array}{c} {\ddot{u}}_{L}={\ddot{u}}_{s}\;\mathrm{on}\;\Gamma \end{array}$$(4)$$\begin{array}{c} t_{L}=t_{s}\;\mathrm{on}\;\Gamma \end{array}$$
to ensure the continuity of acceleration $\ddot{u}$ and tractions t, as well as enforcing Dirichlet and Neumann boundary conditions. The variational formulation of the dynamic equilibrium in Equation (Equation 1) can be described for both $\Omega_{L}$ and $\Omega_{s}$ as the following:(5)$$\begin{array}{c} \int_{\Omega_{L}}\rho_{L}{\ddot{u}}_{L}\cdot \delta \dot{u}d\Omega =\int_{\Gamma_{L}}t_{L}\cdot \delta \dot{u}d\Gamma -\int_{\Omega_{L}}\sigma_{L}:\delta D_{L}d\Omega +\int_{\Omega_{L}}\rho_{L}b_{L}\cdot \delta \dot{u}d\Omega \end{array}$$(6)$$\begin{array}{c} \int_{\Omega_{s}}\rho_{s}{\ddot{u}}_{s}\cdot \delta \dot{u}d\Omega =\int_{\Gamma_{s}}t_{s}\cdot \delta \dot{u}d\Gamma -\int_{\Omega_{s}}\sigma_{s}:\delta D_{s}d\Omega +\int_{\Omega_{s}}\rho_{s}b_{s}\cdot \delta \dot{u}d\Omega \end{array}$$
where we denote a variational velocity $\delta \dot{u}\in V_{0}$, in a space $V_{0}$ where $\forall \delta \dot{u}\in H^{1}\left(\Omega_{i}\right)$, and D as the rate of deformation. A discrete approximation of the variational form can be reduced to the following ordinary differential equations:(7)$$\begin{array}{c} M_{L}{\ddot{u}}_{L}=f_{L}^{\mathrm{ext}}-f_{L}^{\mathrm{int}};\;M_{s}{\ddot{u}}_{s}=f_{s}^{\mathrm{ext}}-f_{s}^{\mathrm{int}} \end{array}$$
where for subdomains $\Omega_{L}$ and $\Omega_{s}$, we sum, over a number of finite elements, $N_{L}$ and $N_{s}$, in each subdomain:(8)$$\begin{array}{c} M_{L}=\sum_{e=1}^{N_{L}}\int_{\Omega_{e}}\rho_{L}N^{T}Nd\Omega_{e};\;M_{s}=\sum_{e=1}^{N_{s}}\int_{\Omega_{e}}\rho_{s}N^{T}Nd\Omega_{e} \end{array}$$(9)$$\begin{array}{c} f_{L}^{\mathrm{ext}}=\sum_{e=1}^{N_{L}}\int_{\Omega_{e}}N^{T}t_{L}d\Omega_{e};\;f_{s}^{\mathrm{ext}}=\sum_{e=1}^{N_{s}}\int_{\Omega_{e}}N^{T}t_{s}d\Omega_{e} \end{array}$$(10)$$\begin{array}{c} f_{L}^{\mathrm{int}}=\sum_{e=1}^{N_{L}}\int_{\Omega_{e}}B^{T}\sigma_{L}d\Omega_{e}\;f_{s}^{\mathrm{int}}=\sum_{e=1}^{N_{s}}\int_{\Omega_{e}}B^{T}\sigma_{s}d\Omega_{e} \end{array}$$

The Lagrangian shape functions are represented by N, with their derivatives denoted as B. As is common within the explicit finite element method, M is lumped for each subdomain, with $f^{\mathrm{ext}}$ and $f^{\mathrm{int}}$ computed with vectors too. We elect to use the leapfrog time integration scheme to step through time, staggering the solution of each kinematic quantity such that:(11)$$\begin{array}{c} {\ddot{u}}_{L}^{n}=M_{L}^{-1}\left(f_{L}^{\mathrm{ext}}-f_{L}^{\mathrm{int}}\right);\;{\ddot{u}}_{s}^{n}=M_{s}^{-1}\left(f_{s}^{\mathrm{ext}}-f_{s}^{\mathrm{int}}\right) \end{array}$$(12)$$\begin{array}{c} {\dot{u}}_{L}^{n+1/2}={\dot{u}}_{L}^{n-1/2}+{\ddot{u}}_{L}^{n}\cdot \Delta t_{L};\;{\dot{u}}_{s}^{n+1/2}={\dot{u}}_{s}^{n-1/2}+{\ddot{u}}_{s}^{n}\cdot \Delta t_{s} \end{array}$$(13)$$\begin{array}{c} u_{L}^{n+1}=u_{L}^{n}+{\dot{u}}_{L}^{n+1/2}\cdot \Delta t_{L};\;u_{s}^{n+1}=u_{s}^{n}+{\dot{u}}_{s}^{n+1/2}\cdot \Delta t_{s} \end{array}$$
noting that a diagonal mass matrix allows for a direct computation of acceleration. Velocity $\dot{u}$ is computed on the half time step, with displacement u found for each full time step. Next, we summarise the temporal coupling, enabled by multi-time-step (MTS) integration.

## 3. Multi-Time-Step Integration

Multi-time stepping enables partitioned subdomains $\Omega_{L}$ and $\Omega_{s}$ to integrate with $\Delta t_{L}$ and $\Delta t_{s}$, respectively. However, to allow for this difference, special attention must be given to the solution of the interface Γ. Crucially, the conditional stability of explicit methods requires an element’s time step to obey the CFL condition for a linear undamped system as follows:(14)$$\Delta t_{C}=\frac{2}{\omega_{C}}\leq \min_{e}\left(\frac{h_{e}}{c_{e}}\right)$$

Here, we represent $\Delta t_{C}$ as the critical time step, $\omega_{C}$ as the maximum eigenfrequency, $h_{e}$ as the characteristic length of an element *e* and $c_{e}$ the dilatational (longitudinal) wave speed.

### 3.1. Salient Multi-Time-Stepping Features

The asynchronous integration is enabled with three important groups of computations. The first is the explicit computation of the acceleration on the interface Γ:(15)$$\begin{array}{c} M_{\Gamma }=\left(C_{s}^{T}M_{s}C_{s}\right)+\left(C_{L}^{T}M_{L}C_{L}\right) \end{array}$$(16)$$\begin{array}{c} f_{\Gamma }^{\mathrm{int}}=C_{s}^{T}f_{s}^{\mathrm{int}}+C_{L}^{T}f_{L}^{\mathrm{int}};\;f_{\Gamma }^{\mathrm{ext}}=C_{s}^{T}f_{s}^{\mathrm{ext}}+C_{L}^{T}f_{L}^{\mathrm{ext}} \end{array}$$

We define indicator matrices (vectors in 1-D) for each subdomain C to identify the degrees of freedom on the interface of subdomains with dimensions $C_{i}\in R^{N_{n_{i}}\times N_{\Gamma }}$ for nodal number N. These summations provide the ingredients for computing the interface acceleration:(17)$$\begin{array}{c} {\ddot{u}}_{\Gamma }=M_{\Gamma }^{-1}\left(f_{\Gamma }^{\mathrm{ext}}-f_{\Gamma }^{\mathrm{int}}\right) \end{array}$$
where we compute ${\ddot{u}}_{\Gamma }$ at each large time step $\Delta t_{L}$. The integration of the subdomains is controlled by the definition of the time-step ratios. Suppose the two subdomains begin at a similar point in time $t_{L}^{N}=t_{s}^{n}$, where *N* and *n* are the small and large steps, respectively. The time after the maximum stable integration step (governed by the CFL condition) on each subdomain is referred to as the *trial time*
$t_{T}$, such that:(18)$$\begin{array}{c} t_{TL}^{N+1}=t_{L}+\Delta t_{CL};\;t_{Ts}^{n+k}=t_{s}+\Delta t_{Cs} \end{array}$$
where for every $\Delta t_{L}$, *k* small time steps elapse since the last point in time where subdomains are equal in time. Now we can define the *current time-step ratio*, $t_{ratio}^{n+k}$, and *next time-step ratio*, $t_{ratio}^{n+k+1}$ for the advancement of $\Omega_{s}$ with:(19)$$\begin{array}{c} t_{ratio}^{n+k}=\frac{t_{s}^{n+k}-t_{L}^{N}}{t_{TL}^{N+1}-t_{L}^{N}};\;t_{ratio}^{n+k+1}=\frac{\left(t_{s}^{n+k}+\Delta t_{s}^{n+k}\right)-t_{L}^{N}}{t_{TL}^{N+1}-t_{L}^{N}} \end{array}$$

Starting from each common time step with the integration of $\Omega_{s}$, the number of small time steps $\Delta t_{s}$ is determined by the evaluation of time-step ratios $t_{ratio}^{n+k}$ and $t_{ratio}^{n+k+1}$. If the condition of $t_{ratio}^{n+k+1}\leq 1$ or ($t_{ratio}^{n+k}\leq 1$ and $t_{ratio}^{n+k+1}\leq 1$) is satisfied, further steps on $\Delta t_{s}$ can proceed before integrating $\Omega_{L}$ over $\Delta t_{L}$. As a consequence of subdomains integrating over their own respective time step, we ensure that the proposed method still finds a common time between all subdomains. Following the small trial time $t_{Ts}$ exceeding the large trial time $t_{TL}$, the method computes two additional ratios $\alpha_{L}$ and $\alpha_{s}$. These denote the *reduction factors* required to maintain the subdomains in synchronisation where 0≤α≤1. Hence:(20)$$\begin{array}{c} \alpha_{L}=1-\frac{\left(t_{TL}^{N}-t_{s}^{n+k}\right)}{\left(t_{TL}^{N+1}-t_{L}^{N}\right)};\;\alpha_{s}=1-\frac{\left(t_{Ts}^{n+k}-t_{TL}^{N+1}\right)}{\left(t_{Ts}^{n+k+1}-t_{s}^{n+k}\right)} \end{array}$$

Through computing α on both subdomains, we can compare reduction factors, such that:(21)$$\begin{array}{c} t_{L}^{N+1}=t_{s}^{n+k}=\left\{\begin{array}{c} t_{s}^{n}+\sum_{k=0}^{k-1}\Delta t_{Cs}^{n+k}+\alpha_{s}\cdot \Delta t_{Cs}^{n+k},\;\alpha_{s}>\alpha_{L}\\ t_{s}^{n}+\alpha_{L}\cdot \Delta t_{CL}^{N},\;\alpha_{L}\geq \alpha_{s} \end{array}\right. \end{array}$$
where we look to reduce the time step of the subdomain closest to the CFL condition. In the following section, we summarise the multi-time-stepping algorithm, with each of these key features, as well as its implementation.

### 3.2. Summary of Temporal Algorithm

We provide an overview of the method required to integrate $\Omega_{L}$ and $\Omega_{s}$ with $\Delta t_{L}$ and $\Delta t_{s}$. It shows a single large step *N*, exemplifying each of the features mentioned above. Note that $TOL=1\times 10^{-6}$ is used for when $\Delta t_{L}\approx \Delta t_{s}$, and computation of $M_{\Gamma }$ in Equation (Equation 15) is only required at t=0 for a mass-conserving problem. One full loop of the procedure times $t_{L}=t_{s}$ is followed, with the subdomain synchronised after each large step *N*. The proposed method is implemented in an open-source python code, found in the following repository: https://github.com/kinfungchan/multi-time-step-integration (accessed on 9 December 2024) [35]. It contains re-implementations of methods from the literature for the two-subdomain cases [9,10]. Whilst we depict the case of just two subdomains, the Algorithm 1 can be extended to multiple by processing subdomains as pairs.
**Algorithm 1** Summary of Algorithm for Coupling in Time from *N* to N+1  1:**procedure** a two-subdomain multi-time integration step  2:      **while**
$t_{ratio}^{n+k+1}\leq 1$ or ($t_{ratio}^{n+k}\leq 1$ and $t_{ratio}^{n+k+1}\leq 1+\mathrm{TOL}$) **do**  3:         Integrate subdomain $\Omega_{s}$ with ${\ddot{u}}_{\Gamma }$ and compute force vectors $f_{s}^{\mathrm{int}},f_{s}^{\mathrm{ext}}$ over $\Delta t_{s}$  4:         Compute trial times $t_{Ts}^{n+k}$, $t_{TL}^{N+1}$ and time step ratios $t_{ratio}^{n+k}$, $t_{ratio}^{n+k+1}$  5:      Compute time step reduction factors $\alpha_{L},\alpha_{s}$  6:      **if**
$\alpha_{L}\geq \alpha_{s}$
**then**  7:         $\Delta t_{L}=\alpha_{L}\cdot \Delta t_{L}$  8:      **else**  9:         $\Delta t_{s}=\alpha_{s}\cdot \Delta t_{s}$10:         Integrate subdomain $\Omega_{s}$ with ${\ddot{u}}_{\Gamma }$ and recompute force vectors $f_{s}^{\mathrm{int}},f_{s}^{\mathrm{ext}}$ over $\Delta t_{s}$11:         Recompute trial times $t_{Ts}^{n+k}$, $t_{TL}^{N+1}$ and time step ratios $t_{ratio}^{n+k}$, $t_{ratio}^{n+k+1}$12:      Integrate subdomain $\Omega_{L}$ with ${\ddot{u}}_{\Gamma }$ and compute force vectors $f_{L}^{\mathrm{int}},f_{L}^{\mathrm{ext}}$ over $\Delta t_{L}$13:      Compute interface acceleration ${\ddot{u}}_{\Gamma }$ with Equations (Equation 15)–(Equation 17) for next time step14:      Recompute trial times $t_{Ts}^{n+k}$, $t_{TL}^{N+1}$ and time step ratios $t_{ratio}^{n+k}$, $t_{ratio}^{n+k+1}$

### 3.3. Numerical Examples in Time

We present a numerical example in 1-D, with the elastic wave propagation through a heterogeneous bar. Suppose the domain Ω is split into two subdomains $\Omega_{L}$ and $\Omega_{s}$ of similar discretisation, with isotropic elastic properties of $E_{L}=207$ GPa, $E_{s}=1000$ GPa and $\rho_{L}=\rho_{s}=7.83\times 10^{-6}$ kgmm^−3^. These material properties result in a non-integer m=2.19, where the time-step ratio is solely driven by the dissimilar material properties. Figure 2 depicts the bar configuration. The velocity boundary condition is applied to $\Omega_{L}$ at x=0 with $\dot{u}\left(t\right)=0.01\sin\left(2\pi \omega_{L}t\right)$ ms^−1^, where we define a half sine wave with a frequency of $\omega_{L}={\left(125(\left(7.83\times 10^{-6}\right)/207)\right)}^{-1}$ rads^−1^. The difference in material impedance results in a portion of the incident wave being transmitted and the remainder reflected in the opposite direction.

In Figure 3, a comparison between the coupled solution ($\Omega_{L}$ with $\Omega_{s}$) and the monolithic (single-time-step) solution is presented at four separate time stamps. The multi-time-step solution solves ${\dot{u}}_{L}$ and ${\dot{u}}_{s}$ over $\Delta t_{L}$ and $\Delta t_{s}$, whereas ${\dot{u}}_{mono}$ is limited by $\Delta t_{s}$. Consequently, for m=2.19, our method reduces the number of integration steps on $\Omega_{L}$ by half. From prescription of the full wave at t1, through to the transmission and reflection of the wave at t4, the MTS solution aligns very well with the single-time-step solution, despite halving the number of large time steps. This reduction in computational effort is even more prominent for highly heterogeneous configurations, as well as variance in spatial discretisation.

For the above simulation, we also compute the energy components, at each small time step, of each subdomain with the following:(22)$$\begin{array}{c} W_{\mathrm{ext}}^{n+1}=W_{\mathrm{ext}}^{n}+\frac{\Delta t_{i}^{n+1/2}}{2}{\left({\dot{u}}_{i}^{n+1/2}\right)}^{T}\left(f_{\mathrm{ext}}^{n}+f_{\mathrm{ext}}^{n+1}\right)=W_{\mathrm{ext}}^{n}+\frac{1}{2}\Delta u_{i}^{T}\left(f_{\mathrm{ext}}^{n}+f_{\mathrm{ext}}^{n+1}\right) \end{array}$$(23)$$\begin{array}{c} W_{\mathrm{int}}^{n+1}=W_{\mathrm{int}}^{n}+\frac{\Delta t_{i}^{n+1/2}}{2}{\left({\dot{u}}_{i}^{n+1/2}\right)}^{T}\left(f_{\mathrm{int}}^{n}+f_{\mathrm{int}}^{n+1}\right)=W_{\mathrm{int}}^{n}+\frac{1}{2}\Delta u_{i}^{T}\left(f_{\mathrm{int}}^{n}+f_{\mathrm{int}}^{n+1}\right) \end{array}$$(24)$$\begin{array}{c} W_{\mathrm{kin}}^{n}=\frac{1}{2}{\left({\dot{u}}_{i}^{n+1/2}\right)}^{T}M{\dot{u}}_{i}^{n+1/2} \end{array}$$
where *n* can be interchanged with *N* when evaluating $\Omega_{L}$. The balance of energy can be evaluated in a similar way to the works of Neal and Belytschko [3], with the following:(25)$$\begin{array}{c} |W_{\mathrm{ext}}-W_{\mathrm{int}}+W_{\mathrm{kin}}\left|\leq |\right|W_{bal}\left|\right| \end{array}$$

In Figure 4, we show each of the components of energy and its overall balance. For both monolithic and multi-time-step solutions, a smooth transition of energy is observed as the wave interacts with Γ. Remarkably, as the temporal coupling is enforced, the $W_{bal}$ for both $\Omega_{L}$ and $\Omega_{s}$ is of the order $1\times 10^{-13}$ kNmm, significantly below each of the components at $1\times 10^{-8}$ kNmm. This numerical example captures the propagation of a smooth pulse; however, severe loading cases, highly heterogeneous domains, and 3-D problems, with further details are provided in the following work [35].

## 4. Solving Non-Matching Meshes

The problem of non-matching meshes is commonly found when simulating the dynamical behaviour of solids. We present an algorithm, combined with multi-time stepping, that relaxes the constraint of these conforming nodes, allowing for a coarser representation of a subdomain to be utilised, hence reducing the computational overhead.

### 4.1. Combined Spatial and Temporal Coupling

The following section follows on from the governing equations defined in Section 2; however, we now allow for the non-overlapping interface $\Gamma =\Gamma_{L}\cap \Gamma_{s}$ to consist of two non-matching spatial discretisations. Their compatibility is maintained such that:(26)$$\begin{array}{c} {\ddot{u}}_{\Gamma L}\left(C_{L}x_{L}\right)={\ddot{u}}_{\Gamma s}\left(C_{s}x_{L}\right)={\ddot{u}}_{\Gamma }\left(x_{\Gamma }\right) \end{array}$$(27)$$\begin{array}{c} t_{\Gamma L}\left(C_{L}x_{L}\right)=t_{\Gamma s}\left(C_{s}x_{s}\right)=t_{\Gamma }\left(x_{\Gamma }\right) \end{array}$$
where positions use an incidence C matrix, such that $C_{L}x_{L}\in \Gamma_{L}$ and $C_{s}x_{s}\in \Gamma_{s}$ for each subdomain, and the externally assembled interface contains $x_{\Gamma }\in \Gamma$ to describe the common boundary between the subdomains. The total virtual power can be summated for two subdomains to give $\delta P=\delta P_{L}+\delta P_{s}+\delta P_{\Gamma }$, where a general form is obtained:(28)$$\begin{array}{cc} \delta P & =\delta {\dot{u}}_{L}^{T}\left\{f_{L}^{\mathrm{int}}-f_{L}^{\mathrm{ext}}+M_{L}{\ddot{u}}_{L}+N_{L}^{T}f_{\Gamma L}\right\}+\delta {\dot{u}}_{s}^{T}\left\{f_{s}^{\mathrm{int}}-f_{s}^{\mathrm{ext}}+M_{s}{\ddot{u}}_{s}+N_{s}^{T}f_{\Gamma s}\right\}\\ & +\delta f_{\Gamma L}^{T}\left\{N_{L}^{T}{\dot{u}}_{\Gamma L}-L_{L}^{T}{\dot{u}}_{\Gamma }\right\}+\delta f_{\Gamma s}^{T}\left\{N_{s}^{T}{\dot{u}}_{\Gamma s}-L_{s}^{T}{\dot{u}}_{\Gamma }\right\}-\delta {\dot{u}}_{\Gamma }^{T}\left\{N_{L}^{T}f_{\Gamma L}+N_{s}^{T}f_{\Gamma s}\right\} \end{array}$$
where variations in velocity coupling force on the interface $\dot{u}$ and $f_{\Gamma i}$ are accounted for. $N_{i}^{T}$ and $L_{i}^{T}$ are interpolation (or prolongation) and incidence operators (Γ to $\Omega_{i}$), respectively. To map to the interface, we describe this spatial coupling operator $N_{i}$ in more detail:(29)$$\begin{array}{c} N_{i}\in R^{N_{\Gamma i}\times N_{\Gamma }} \end{array}$$
with dimensions determined by $N_{\Gamma i}$ and $N_{\Gamma }$, as the number of nodes on the interface of the subdomain $\Omega_{i}$ and the number of nodes on the interface Γ, respectively. Therefore, $N_{\Gamma }$ interpolates using Lagrangian shape functions for the two subdomains $\Omega_{L}$ and $\Omega_{s}$:(30)$$\begin{array}{c} N_{\Gamma }\left(x\right)=\delta \left(x_{\Gamma }-x_{\Gamma i}\right),\;\mathrm{for}\;i=L,s \end{array}$$
where δ is viewed as a dirac Delta function for coincident nodes. It is convenient to define a restriction operator $R_{i}\in R^{N_{\Gamma }\times N_{\Gamma i}}$ as the transpose of $N_{i}$, to map both forces and mass from subdomain interfaces $\Gamma_{L}$ and $\Gamma_{s}$ onto Γ. The summation on the interfaces now becomes:(31)$$\begin{array}{c} M_{\Gamma }=R_{L}C_{L}^{T}M_{L}+R_{s}C_{s}^{T}M_{s}=R_{L}M_{\Gamma L}+R_{s}M_{\Gamma s} \end{array}$$(32)$$\begin{array}{c} f_{\Gamma }^{\mathrm{int}}=R_{L}C_{L}^{T}f_{L}^{\mathrm{int}}+R_{s}C_{s}^{T}f_{s}^{\mathrm{int}}=R_{L}f_{\Gamma L}^{\mathrm{int}}+R_{s}f_{\Gamma s}^{\mathrm{int}} \end{array}$$(33)$$\begin{array}{c} f_{\Gamma }^{\mathrm{ext}}=R_{L}C_{L}^{T}f_{L}^{\mathrm{ext}}+R_{s}C_{s}^{T}f_{s}^{\mathrm{ext}}=R_{L}f_{\Gamma L}^{\mathrm{ext}}+R_{s}f_{\Gamma s}^{\mathrm{ext}} \end{array}$$
where we compute mass $M_{\Gamma }$, internal force $f_{\Gamma }^{\mathrm{int}}$ and external force $f_{\Gamma }^{\mathrm{ext}}$ to allow for the explicit computation of ${\ddot{u}}_{\Gamma }$ in Equation (Equation 17) to be recalled. These operators are analogous to concepts in multigrid methods and localised Lagrange multipliers (LLMs) [36,37]. Subsequently, we map ${\ddot{u}}_{\Gamma }$ from Γ, back to the subdomains’ interfaces:(34)$$\begin{array}{c} {\ddot{u}}_{\Gamma L}=N_{L}{\ddot{u}}_{\Gamma };\;{\ddot{u}}_{\Gamma s}=N_{s}{\ddot{u}}_{\Gamma } \end{array}$$

We illustrate a non-matching mesh in Figure 5 and compute its operators through exemplifying a linear isoparametric mapping in 2-D, where Γ is discretised with line elements to depict the simplicity of this coupling.

For the interpolation matrix, we elucidate that from Γ to $\Gamma_{L}$ the mapping is simply one-to-one and $N_{L}$ will always take the form of an identity matrix $\Omega_{L}$, whereas $N_{s}$ requires computation of the shape functions:(35)$$\begin{array}{c} N_{L}=\left[\begin{array}{ccc} 1 & 0 & 0\\ 0 & 1 & 0\\ 0 & 0 & 1 \end{array}\right];\;N_{s}=\left[\begin{array}{ccc} 1 & 0 & 0\\ 1/3 & 2/3 & 0\\ 0 & 2/3 & 1/3\\ 0 & 0 & 1 \end{array}\right];\;R_{L}=N_{L}^{T};\;\;R_{s}=N_{s}^{T} \end{array}$$
where, for this example case, it can also be shown that the restriction is the transpose of the interpolation. The interface Γ is assumed to share the same geometrical description on both subdomain interfaces $\Gamma_{L}$ and $\Gamma_{s}$, without overlap or separation. The spatial and temporal methods are combined to give the following Algorithm 2:
**Algorithm 2** Summary of Non-Matching Mesh Algorithm with Multi-Time Stepping  1:**procedure**integrate a two-domain non-matching mesh with MTS  2:      **while**
$t_{ratio}^{n+k+1}\leq 1$ or ($t_{ratio}^{n+k}\leq 1$ and $t_{ratio}^{n+k+1}\leq 1+TOL$) **do**  3:         Compute ${\ddot{u}}_{\Gamma_{s}}$ with operator $N_{s}$ in Equation (Equation 34) on $\Gamma_{s}$  4:         Integrate small domain $\Omega_{s}$ and compute vectors $f_{s}^{\mathrm{int}},f_{s}^{\mathrm{ext}}$  5:         Compute trial times $t_{Ts}^{n+k}$, $t_{TL}^{N+1}$ and time step ratios $t_{ratio}^{n+k}$, $t_{ratio}^{n+k+1}$  6:      Compute time step reduction factors $\alpha_{L},\alpha_{s}$  7:      **if**
$\alpha_{L}\geq \alpha_{s}$
**then**  8:         $\Delta t_{L}=\alpha_{L}\cdot \Delta t_{L}$  9:      **else**10:         $\Delta t_{s}=\alpha_{s}\cdot \Delta t_{s}$11:         Compute ${\ddot{u}}_{\Gamma_{s}}$ with operator $N_{s}$ in Equation (Equation 34) on $\Gamma_{s}$12:         Integrate small domain $\Omega_{s}$ and compute vectors $f_{s}^{\mathrm{int}},f_{s}^{\mathrm{ext}}$13:         Recompute trial times $t_{Ts}^{n+k}$, $t_{TL}^{N+1}$ and time step ratios $t_{ratio}^{n+k}$, $t_{ratio}^{n+k+1}$14:      Compute ${\ddot{u}}_{\Gamma_{L}}$ with operator $N_{L}$ in Equation (Equation 34) on $\Gamma_{L}$15:      Integrate large domain $\Omega_{L}$ and compute vectors $f_{L}^{\mathrm{int}},f_{L}^{\mathrm{ext}}$16:      Summate kinetics with $R_{L},R_{s},C_{L},C_{s}$ to find $M_{\Gamma },f_{\Gamma }^{\mathrm{int}}$ and $f_{\Gamma }^{\mathrm{ext}}$ with Equation (Equation 31)–(Equation 33)17:      Compute trial times $t_{Ts}^{n+k}$, $t_{TL}^{N+1}$ and time step ratios $t_{ratio}^{n+k}$, $t_{ratio}^{n+k+1}$

### 4.2. Numerical Examples in Space and Time

The following numerical study looks to represent the stress gradients prior to fracture in a compact-tension (CT) specimen test, utilising a similar geometry to the literature [38,39]. Figure 6a portrays the geometry modelled in the following example. As the specimen is loaded, stress concentrates about the specimen’s crack tip. We apply a ramped velocity $\dot{u}\left(t\right)$ boundary condition on nodes that create a semicircle for upper and lower pins, as shown in Figure 6b, with a maximum magnitude of 0.2 ms^−1^. Whilst uniformly distributed velocities are applied to each of the nodes in the pins, to replicate the contact pressure on the pins, methods such as those applied by Triclot et al. should be considered [40]. Material properties are similar to alumina with E=370 GPa, $\rho =3.9\times 10^{-6}$ kgmm^−3^ and Poisson’s ratio ν=0.22. We model the CT specimen with three simulations: one using a fine mesh throughout the entire domain, one coupling a coarse $\Omega_{L}$ and fine $\Omega_{s}$ mesh with a single Δt, and another with $\Omega_{L}$ and $\Omega_{s}$ integrating with multiple time steps $\Delta t_{L}$ and $\Delta t_{s}$, respectively. Structured meshes are used in all cases where an element size of 0.58 mm for fine and 1 mm for coarse. All simulations use Co=0.5, running for a maximum of $t_{final}=0.02$ ms.

In Figure 7 and Figure 8, we plot the stress contours $\sigma_{\mathrm{yy}}$ at t=0.01408 ms and t=0.01966 ms, respectively, where the stress concentration can be visualised at the crack tip of the specimen. Figure 7a and Figure 8a capture the reference mesh, whilst Figure 7b and Figure 8b capture the spatially coupled mesh on a single time step $\Delta t_{s}$. Figure 8 accurately predicts maximum stresses of 0.0396 GPa vs. 0.0410 GPa for reference against non-matching mesh.

Further quantification on the performance of the non-matching mesh algorithm can be made with the evaluation of stress intensity factors, near the crack tip. Utilising the stress extrapolation method, stress values in the vicinity of the crack tip can be used to simultaneously solve for in-plane $K_{I}$ and shear $K_{II}$ factors [41,42]. In rectangular coordinates, these are given as follows:(36)$$\begin{array}{c} \sigma_{yy}=\frac{K_{I}}{\sqrt{2\pi r}}\cos\frac{\theta }{2}\left(1-\sin\frac{\theta }{2}\sin\frac{3\theta }{2}\right)-\frac{K_{II}}{\sqrt{2\pi r}}\sin\frac{\theta }{2}\left(2+\cos\frac{\theta }{2}\cos\frac{3\theta }{2}\right) \end{array}$$(37)$$\begin{array}{c} \sigma_{xx}=\frac{K_{I}}{\sqrt{2\pi r}}\cos\frac{\theta }{2}\left(1+\sin\frac{\theta }{2}\sin\frac{3\theta }{2}\right)+\frac{K_{II}}{\sqrt{2\pi r}}\left(\cos\frac{\theta }{2}\sin\frac{\theta }{2}\cos\frac{3\theta }{2}\right) \end{array}$$
where *r* represents the radial distance from the crack, and θ the angle relative to the crack plane. Whilst J-integral and virtual crack closure techniques (VCCTs) have been developed for more accurately evaluating *K*, here the consistency in the method for both reference and coupled solution proves a sufficient comparison [43,44]. The stress $\sigma_{xx}$ and $\sigma_{yy}$ in the nearest element to the crack correspond to Figure 9a where strong alignment between spatially coupled and reference solution can be observed. In Figure 9b, the analysis computes $K_{I}$ and $K_{II}$ with $r^{\mathrm{ref}.}=0.1294$ mm, $\theta^{\mathrm{ref}.}=1.040^{\circ }$, $r^{\mathrm{coup}.}=0.1320$ mm and $\theta^{\mathrm{coup}.}=1.015^{\circ }$ with the slight variation in the spatial discretisation of $\Omega_{s}$ and summarising in Table 1. $K_{I}$ and $K_{II}$ compute a difference of 3.5% and 6.3% in value when comparing uncoupled and coupled solutions. This is deemed reasonable granted the approximation of the stress extrapolation method. For future comparisons, the contribution of multiple Gauss integration points could be considered with the aforementioned J-integral or VCCT methods.

Now with multi-time stepping, $\Omega_{L}$ and $\Omega_{s}$ step with $\Delta t_{L}=3.082\times 10^{-5}$ ms and $\Delta t_{s}=5.315\times 10^{-6}$ ms, producing a time-step ratio m=5.80. Through combining spatial and temporal coupling, we observe similar (<$1\times 10^{-6}$ GPa) results in $\sigma_{\mathrm{yy}}$ distribution, as seen in Figure 10a. The difference in the coupled simulations is captured via Figure 10b, where the root mean square error (RMSE) is plotted. The stress $\sigma_{\mathrm{yy}}$ is linearly interpolated for the non-matching mesh to allow for comparison of stress on the coarse mesh’s Gauss points. The couplings capture nearly the entire specimen within a <5% error. Considerable error is located around the circumference of the pins, where the chequerboard pattern indicates potential hourglassing in $\Omega_{L}$. To mitigate such issues, higher order or fully integrated elements could be used. However, in the presented results, we note that no hourglass control methods have been applied for fair comparison of the reference and coupled solutions. The other portion of error can be seen in the far right of the specimen in Figure 10; however, like the pins, these Gauss points reside far from the area of interest at the crack tip. To avoid such errors, a smaller element size would be required in $\Omega_{L}$; however, this raises the trade-off between accuracy and computational efficiency. As per the temporal coupling, an energy balance in each of the two subdomains was achieved $W_{bal,i}<\times 10^{-12}$ kNmm, orders of magnitude below individual components of energy. The computational efficiency is summarised with speedup achieved utilising both couplings, as presented in Table 2. Whilst a modest speedup is achieved with non-matching meshes, an even larger efficiency is found with coupling in both space and time. Considering that coupling with non-matching meshes and combined coupling in space and time result in similar accuracy, the addition of multi-time stepping to these simulations seems obvious.

## 5. Conclusions

We present coupling methods for the dynamic modelling of solids with explicit finite elements, temporally and spatially. When modelling composites, constituents have varying dilatational wave speeds, hence different time steps. Integrating over the smallest time step can prove highly inefficient, hence the need for multi-time stepping. Our method allows partitions of a domain to solve with their own respective time step, regardless of time-step ratio, hence reducing computational overhead. The method avoids solving a system of equations on the interface, unlike many of the methods that employ Lagrange multipliers. The stability of the method is assessed through evaluation of the subdomains’ energy balance. Very-high-frequency content or variations in motion below the time step of elements on the interface could result in spurious oscillations being generated. This potential limitation promotes the development of adaptive multi-time-stepping schemes that maintain the stability of the interface, withstanding high-frequency stress waves.

The addition of the coupling in space solves the issue of non-matching meshes so that small element sizes are only required in regions of interest. Coupling operators are easily implemented, without increasing the degrees of freedom on the interface. The method avoids the storage of large sparse matrices, reducing computational memory requirements. Ongoing work addresses the spatial coupling with quadrilateral non-matching interfaces in 3-D domains [45]. Numerical examples capture an increase in efficiency with stress wave propagation in a heterogeneous bar, and the modelling of a compact-tension specimen. Both couplings in time and space reduce computational runtimes when compared to their monolithic simulation, especially when combined. Limitations that concern the combination of spatial and temporal coupling include the computation of operators N and R solely at t=0. Whilst this suffices for the benchmarks shown, for larger deformations this assumption is likely to require further development. Geometric representations of a non-matching Γ that are non-planar have still yet to be explored. This proves an important topic as these couplings are applied to real-world multi-scale problems.

Future work looks at the coupling between macro- and meso-scale meshes [46,47,48], with adaptivity a clear necessity for these multi-scale couplings [49]. Whilst linear elasticity is a fair assumption for the rates of deformation demonstrated, other constitutive models should be investigated to evaluate the performance of the couplings, with further reductions in time steps and element distortion. In parallel, experimental fields that require efficiently capturing wave propagation with explicit finite elements are widespread [50,51]. Other coupling opportunities are plentiful when considering dynamic applications; the modelling of contact [14,16], composite fractures [38,39], fluid–structure interactions [13,27], and other impact engineering scenarios are just a few worth mentioning.

## Figures and Tables

**Figure 1 materials-18-01080-f001:**
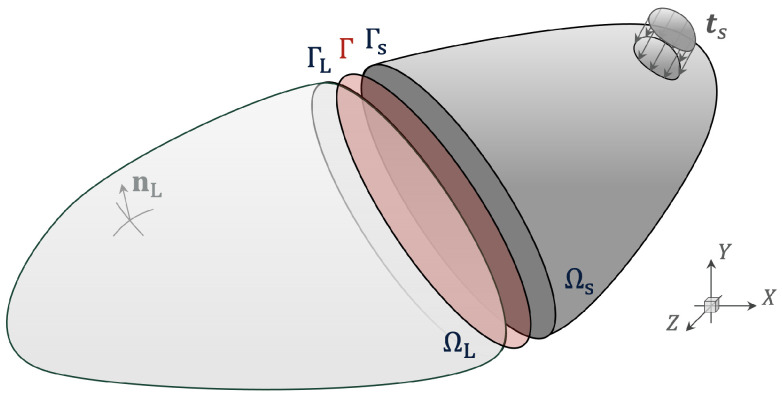
A 3-D domain Ω decomposed into $\Omega_{L}$ and $\Omega_{s}$ where $\Gamma_{L}$ and $\Gamma_{s}$ are coupling interfaces to be externally resolved on Γ. Normal vector and tractions are visualised on $\Omega_{L}$ and $\Omega_{s}$, respectively.

**Figure 2 materials-18-01080-f002:**
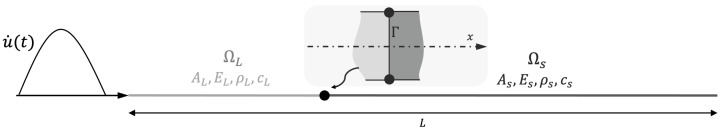
A one-dimensional heterogeneous domain Ω split into a large subdomain $\Omega_{L}$ and a small subdomain $\Omega_{s}$, solved with $\Delta t_{L}$ and $\Delta t_{s}$, respectively, of length $L_{L}=125$ mm and $L_{s}-250$ mm. The problem assumes uni-axial motion with Poisson’s ratio ν=0. A compressive half sine velocity boundary condition is applied from $\Omega_{L}$.

**Figure 3 materials-18-01080-f003:**
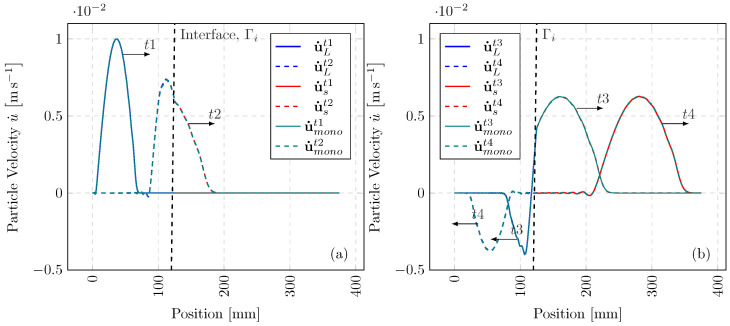
Axial wave propagation in a heterogeneous bar: (**a**)—boundary condition at t1=0.03363 ms and initial transmission at t2=0.04424 ms of the stress wave; (**b**)—transmission and reflection of the stress wave at t3=0.01323 ms and t4=0.02920 ms.

**Figure 4 materials-18-01080-f004:**
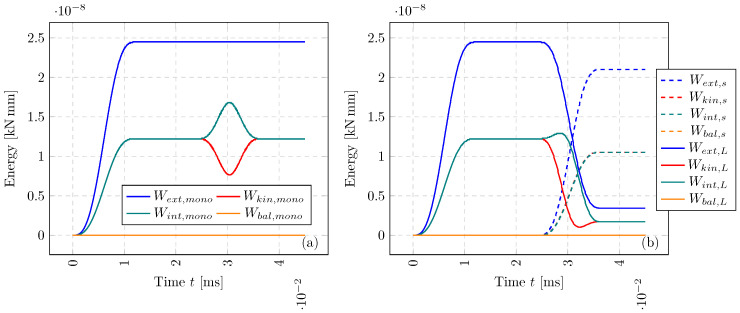
Energy balance for the axial wave propagation problem: (**a**)—monolithic (single $\Delta t_{mono}$) simulation system energy component history and balance with Equations (Equation 22)–(Equation 25), (**b**)—multi-time-stepping ($\Delta t_{L}$ and $\Delta t_{s}$) simulation energy balance.

**Figure 5 materials-18-01080-f005:**
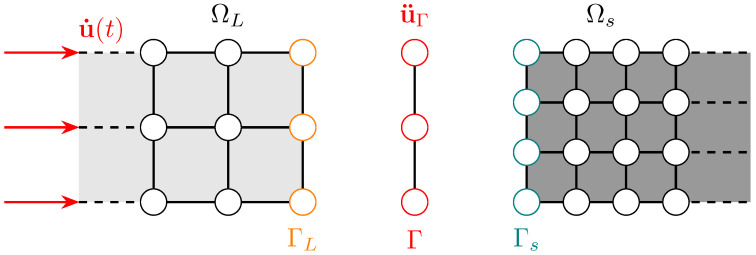
A non-matching benchmark highlighting the interface with differently discretised subdomains partitioned with the large $\Gamma_{L}$, small $\Gamma_{s}$ and externally meshed interface Γ.

**Figure 6 materials-18-01080-f006:**
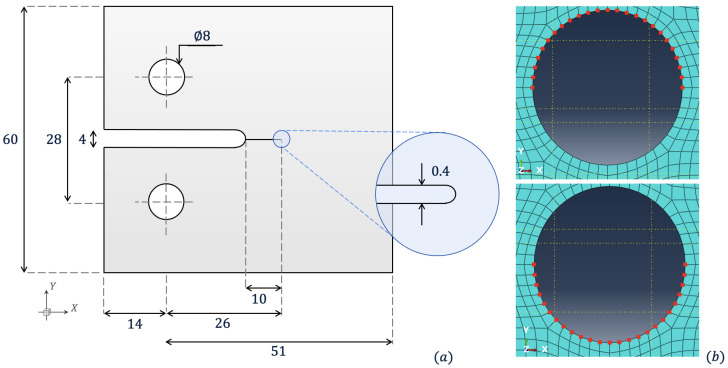
(**a**)—Diagram of compact-tension specimen with dimensions in [mm], as seen in Sommer et al. [39]; (**b**)—nodal sets on the meshed subdomain $\Omega_{L}$ for prescribed velocity boundary conditions on top (+ve loading) and bottom (−ve loading) pins.

**Figure 7 materials-18-01080-f007:**
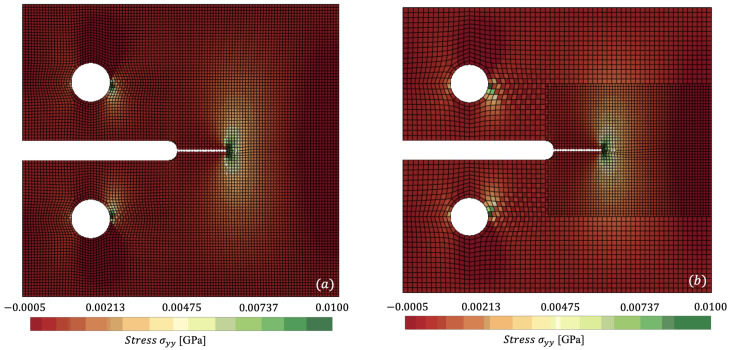
Comparison of $\sigma_{\mathrm{yy}}$ for (**a**) reference (monolithic) versus (**b**) spatially coupled dynamically loaded compact-tension specimen, clipping from −0.0005 to 0.01 GPa at t=0.01408 ms.

**Figure 8 materials-18-01080-f008:**
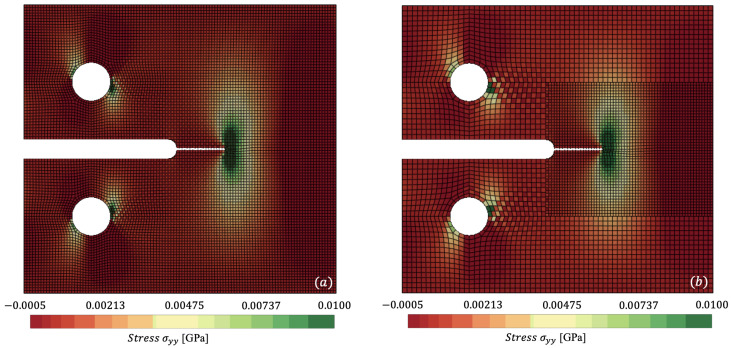
Comparison of $\sigma_{\mathrm{yy}}$ for (**a**) reference (monolithic) versus (**b**) spatially coupled dynamically loaded compact-tension specimen, clipping from −0.0005 to 0.01 GPa at t=0.01966 ms.

**Figure 9 materials-18-01080-f009:**
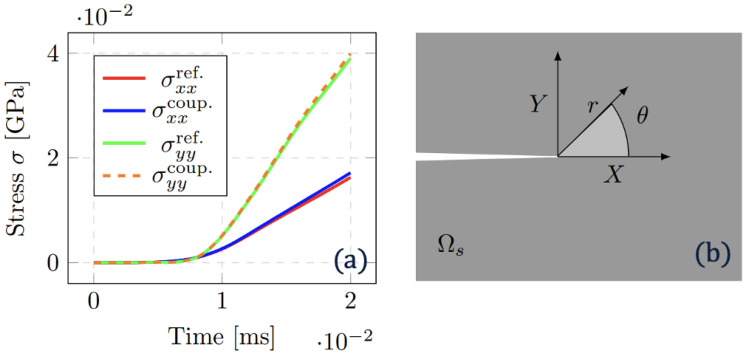
(**a**): Stress evolution with crack tip at coordinates (42.0,30.0) extrapolated from the nearest element comparing the reference (fine mesh) and coupled (coarse and fine mesh) simulation through time. (**b**): The crack tip where radius (*r*) and angle (θ) are used to estimate stress intensity factor (SIF).

**Figure 10 materials-18-01080-f010:**
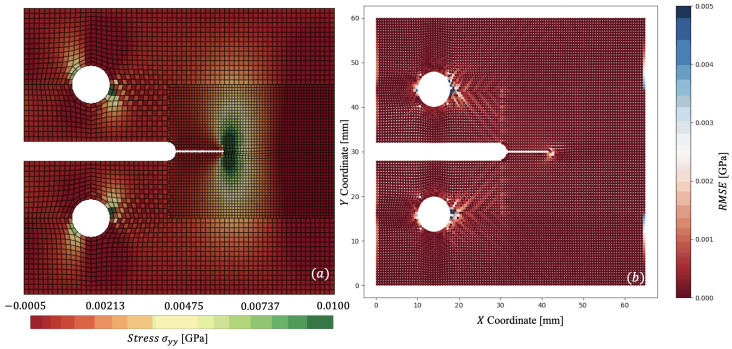
Comparison of $\sigma_{\mathrm{yy}}$ for: (**a**)—combined spatial and temporal coupling (multi-time stepping); (**b**)—RMSE Error of $\sigma_{\mathrm{yy}}$ for dynamically loaded compact-tension specimen at t=0.01966 ms.

**Table 1 materials-18-01080-t001:** Comparison of stress intensity factors $K_{I}$ and $K_{II}$ for the compact-tension specimen test with reference mesh and spatially coupled mesh at final time t=0.02 ms.

Simulation	$K_{I}$ [MPa·$m^{\frac{1}{2}}$]	$K_{\mathrm{II}}$ [MPa·$m^{\frac{1}{2}}$]
Reference (monolithic)	0.02714	−0.002857
Spatial coupling	0.02810	−0.003036

**Table 2 materials-18-01080-t002:** Computational runtimes and speedup vs. reference (single $\Delta t_{s})$, of the dynamically loaded CT specimen. Reference mesh size is 0.58 mm throughout the domain, whereas coupled simulations utilise 0.58 and 1.0 mm for $\Omega_{s}$ and $\Omega_{L}$, respectively. $\Delta t_{L}$ and $\Delta t_{s}$ used for temporally coupled run.

Simulation	Runtime [s]	Speedup
Reference (monolithic)	7428	-
Spatially Coupled	2267	3.27×
Spatially and Temporally Coupled	572	12.98×

## Data Availability

The original contributions presented in the study are included in the article, further inquiries can be directed to the corresponding author.

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
