# Peer review of "Temporal and Spatial Coupling Methods for the Efficient Modelling of Dynamic Solids"

_materials, 2025, doi:10.3390/ma18051080_

Round 1

Reviewer 1 Report

Comments and Suggestions for Authors Dear Authors, Most of what you wrote is reasonable and acceptable. Your work focused on method development and evaluation. However, research requires results, and based on those results, you must conclude something. My recommendation is to rename your conclusion section to discussion or something else. And based on your quantifiable results make conclusions. For example, accuracy, limitations anything. It is good work, but the conclusion and result showcase are a bit weak.

Author Response

Dear Reviewer 1,

Thank you for your time in reviewing this manuscripts, please find attached authors response in which you will find the alterations made to the manuscripts together with the authors response.

Kind Regards

Kin

Reviewer 2 Report

Comments and Suggestions for Authors

The manuscript titled “Temporal and Spatial Coupling Methods for the Efficient Modelling of Dynamic Solids” by Chan, K.F.; et al. is a scientific work where the authors addressed the performance of designed theoretical frameworks to gain calculation time during the dynamic simulation process. The manuscript is generally well-written and this is a topic of growing interest.

However, it exists some points that need to be addressed (please, see them below detailed point-by-point) to improve the scientific quality of the submitted manuscript paper before this article will be consider for its publication in Materials.

1) Keywords. The authors should consider to add “dynamic modelling” in the keyword list.

2) Introduction. “The temporal and spatial discretisation of structural dynamic problems is directly related to the accuracy and computational cost of the explicit finite element method” (lines 12-13). Could the authors provide quantitative data insights according to the economic impact of computing analysis? This will significantly aid the potential readers to better understand the devoted research in this work.

3) Then, it should be also mentioned some experimental fields where the development of more efficient modelling simulaton can be benefited. In this context, the calculation of nanomechanics [1] or frictional forces [2] are two excellent examples where the future implementation of this work could lead to a positive impact.

[1] https://doi.org/10.3390/nano13060963

[2] https://doi.org/10.3390/ma17153828

4) “3. Multi-time Step Integration” (lines 70-143). Did the authors experience any issue related to the appeareance of artificial resonances. This occurs when large time-step discretization excites high-frequency components in the system that could induce an inaccuracy source. A brief statement should be add to discuss about this aspect.

5) “4. Solving Non-Matching Meshes” (lines 144-208). The energy savings linked to this methodology should be also mentioned.

6) “5. Conclusions” (lines 209-277). This section perfectly remarks the most relevant outcomes found by the authors in this work and also the promising future prospectives. It may be advisable to add a brief statement to remark the potential future action lines to pursue the topic covered in this research.

Author Response

Dear Reviewer 2,

Thank you for your time in reviewing this manuscripts, please find attached authors response in which you will find the alterations made to the manuscripts together with the authors response.

Kind Regards

Kin

Reviewer 3 Report

Comments and Suggestions for Authors

The paper presents in a synthetic way a worthy research to improve the mathematics on which finite elements modelling of materials is based. Most finite elements programmes are directed at modelling structures, therefore modelling of materials is welcome, since here numerous assumptions in civil engineering numerical simulation are done. The structure of the paper is suitable for a mathematics paper and includes a suitable example. However, the discussion on how this example performs is not sufficient. The figure and equations are illustrating well the paper. One note is that the algorithm is in a proper programming language but does not detail how this would be employed: as script to an existing FEM software or writing a new software.

Author Response

Dear Reviewer 3,

Thank you for your time in reviewing this manuscripts, please find attached authors response in which you will find the alterations made to the manuscripts together with the authors response.

Kind Regards

Kin
